# Enhanced recombination empowers the detection and mapping of Quantitative Trait Loci
Laia Capilla-Pérez[1], Victor Solier[1], Elodie Gilbault[2], Qichao Lian[1], Manish Goel[1,5], Bruno Huettel[3], Joost J. B. Keurentjes[4], Olivier Loudet[2]✉ & Raphael Mercier[1]✉

Modern plant breeding, such as genomic selection and gene editing, is based on the knowledge of the genetic architecture of desired traits. Quantitative trait loci (QTL) analysis, which combines high throughput phenotyping and genotyping of segregating populations, is a powerful tool to identify these genetic determinants and to decipher the underlying mechanisms. However, meiotic recombination, which shuffles genetic information between generations, is limited: Typically only one to two exchange points, called crossovers, occur between a pair of homologous chromosomes. Here we test the effect on QTL analysis of boosting recombination, by mutating the anti-crossover factors *RECQ4* and *FIGL1* in *Arabidopsis thaliana* full hybrids and lines in which a single chromosome is hybrid. We show that increasing recombination ~6-fold empowers the detection and resolution of QTLs, reaching the gene scale with only a few hundred plants. Further, enhanced recombination unmasks some secondary QTLs undetected under normal recombination. These results show the benefits of enhanced recombination to decipher the genetic bases of traits.

Meiotic recombination reshuffles parental genetic material at each generation, resulting in genetic variability in the offspring. It is exploited both to localize loci with respect to each other in genetic mapping approaches and to mix genetic features from different parents in breeding. Overall, the rather low level of recombination (typically one or two per chromosome per generation[1]) has long been known as a limiting factor for studies such as QTL (Quantitative Trait Locus) mapping, that aim to link complex traits with their genetic architecture. QTL studies have been widely used especially in plants to understand the phenotypical diversity of quantitative traits, its genetic bases, or to guide crop improvement. Trying to improve the power of QTL detection and control the recombination landscape has been the focus of breeding programs for many years[2]. One strategy that has been exploited in mapping populations is increasing the recombination density per line by performing several generations of intercrossing before fixing the recombinant lines (e.g. Advanced-Intercrossed RIL in[3]). However, the increase remains limited, especially considering the work it represents and its complexity. Another strategy has been to genotypically select the lines with more recombination[4], also with limited effect.

Boosting crossovers by manipulating the recombination machinery is an alternative attractive approach to improve mapping. Meiotic crossovers formation initiates with the generation of hundreds of programmed double-strand breaks (DSBs), a minority of them being matured into crossovers, the rest being repaired as non-crossovers[1,5–7]. There are two classes of crossovers, as defined by their biochemical pathways, class I, which accounts for the vast majority of crossovers, and class II which accounts for a small number of events in wild-type meiosis. Several mechanisms limit the number of COs in plants. Three protein complexes limit the formation of class II CO: TOP3/RECQ4AB/RMI1[8–10], FANCM and associated proteins[11–14] and FIGL1/FLIP[15,16]. In parallel, the number of class I crossovers is limited by the dosage of the pro-CO factor HEI10[17], the phosphatase X1[18], and the synaptonemal complex proteins ZYP1/SCEP1/SCEP2[19–21]. Combining mutation in these pathways leads to a cumulative increase in recombination, the highest level of crossovers observed to date in Arabidopsis being obtained by combining the *recq4* and *figl1* mutations[22–24].

In addition to the manipulation of recombination, the availability of specific genetic resources such as Chromosome Substitution Lines (CSLs)[25–27] can also empower QTL detection. These lines are characterized

[1]Max Planck Institute for Plant Breeding Research, MPIPZ, Department of Chromosome Biology, Carl-von-Linné Weg 10, 50829 Cologne, Germany. [2]Université Paris-Saclay, INRAE, AgroParisTech, Institut Jean-Pierre Bourgin (IJPB), 78000 Versailles, France. [3]Max Planck Institute for Plant Breeding Research, MPIPZ, Genome Center, Carl-von-Linné Weg 10, 50829 Cologne, Germany. [4]Laboratory of Genetics, Wageningen University & Research, Droevendaalsesteeg 1, 6708 PB Wageningen, The Netherlands. [5]Present address: Ludwig-Maximilians-Universität München, Fakultät für Biologie, Biozentrum Martinsried, 82152 Planegg-Martinsried, Germany. ✉e-mail: olivier.loudet@inrae.fr; mercier@mpipz.mpg.de

by having four homozygous chromosomes from one accession (the host accession, e.g. Columbia) while just one chromosome is replaced from another accession (the donor, e.g. Landsberg). This permits it to have only one segregating chromosome in hybrids, potentially improving QTL detection by simplifying the genetic and epistasis landscape.

In this work, we have first reanalyzed recombination in *Arabidopsis thaliana recq4 figl1* hybrids using whole genome sequencing and showed that crossovers are boosted ~6-fold compared to wild type. Using these hyper-recombinant segregating populations either in Col/Ler full hybrids or in lines segregating for a single chromosome, we have carried out a QTL analysis on traits relative to the plant shape, color, and transition to flowering. Our results showed that both QTL detection and mapping resolution are strongly improved by enhanced recombination.

## Results

### *recq4ab figl1* enhance recombination in full hybrid and substitution lines

Previous work combining knockouts for different anti-CO pathways showed that the genotype that gives the strongest increase so far, is obtained by mutating *FIGL1* (AT3G27120) together with the two *RECQ4* paralogs *RECQ4A* (AT1G10930) and *RECQ4B* (At1G60930)[23]. Here, we combined the same mutations (Figure S1-S2) into F1-hybrid populations with different chromosomal configurations : (i) The hybrid between the Columbia (Col) and Landsberg (Ler) strains (Fig. 1a) and (ii) a set of chromosome substitution lines (CSLs) in which a single chromosome is segregating (Col/Ler), while the rest of the genome is fixed (Fig. 2). Hereby, we will identify these lines as: The full hybrid population (Fig. 1), CSL_chr2_L, and CSL_chr5_L for the Chromosome Substitution Line with chromosome 2 or chromosome 5 segregating, respectively (and the rest of chromosomes homozygous Ler) (Fig. 2a, b), and CSL_chr4_C and CSL_chr5_C for the lines with chromosome 4 or chromosome 5 segregating, respectively (with the rest of chromosomes being Col) (Fig. 2c, d). For each of these hybrids, populations of F2 plantlets were produced by self-fertilization and whole genome sequenced with Illumina short reads. COs were detected using a sliding window approach as described in Lian et al.[28]. Sequence data were

also used to identify potential aneuploids using relative coverage between chromosomes[29,30].

In Col/Ler full hybrids, the average number of COs per F2 plant was 8.4 +/− 2.3 for the wild type, ranging from 2 to 16, corresponding to 1.7 CO per chromosome and similar to previous reports[28,31]. In *recq4ab figl1* full hybrids the number of CO per F2 was 50.5 +/− 11.8, ranging from 20 to 87, corresponding to a massive increase of six-fold and an average of ten CO per chromosome (Fig. 1b). This confirms the synergistic effect of RECQ4 and FIGL1 in limiting CO formation[23].

Looking at the distribution of COs along the genome in the full hybrid (Fig. 1c), frequencies are increased all along the arms in *recq4ab figl1* compared to the wild type. However, the centromeric regions which are off for recombination in wild type, remain off in *recq4ab figl1*. Further, in regions adjacent to the centromeres, where recombination is high in wild type, CO frequency is not or only slightly increased by the *recq4ab figl1* mutations, suggesting that additional factors limit COs in these regions. A lower increase in CO is also observed around position 23 Mb on chromosome 1, which corresponds to a region of high polymorphism associated with a cluster of NBS-LRR disease-resistance genes.

Next, we observed the effect of mutation *recq4ab figl1* in the hybrid chromosome of the CSL populations. In CSL_Chr2_L and CSL_Chr5_L, chromosome 2 and 5 segregates, respectively, while the rest of the genome is fixed Ler (Fig. 2a, b). In CSL_Chr2_L, the number and distribution of COs on chromosome 2 are indistinguishable from the same chromosome in the full hybrid context, in both wild type (1.5 COs, Mann-Whitney test, *p* = 0.92) and *recq4ab figl1* (8 COs, *p* = 0.83) (Fig. 2e). Concerning chromosome 5, it receives 1.9 COs in the wild-type full hybrid context and 2.2 in the wild-type CSL_Chr5_L (Mann-Whitney test *p* = 0.017), suggesting that a trans factor slightly affects CO number on chromosome 5, with the Col allele favoring COs. As in full hybrid, COs are strongly boosted by *recq4 figl1* in CSL-Chr5_L on chromosome 5, to reach similar levels (12 and 11.4, respectively. *p* = 0.14) (Fig. 2f). Thus, CO frequency can be enhanced in substitution lines, exactly like in full hybrids ( ~ 6 folds).

Strikingly, a different observation was made in CSL_Chr4_C and CSL_Chr5_C, in which chromosome 4 and 5 segregate, respectively, while

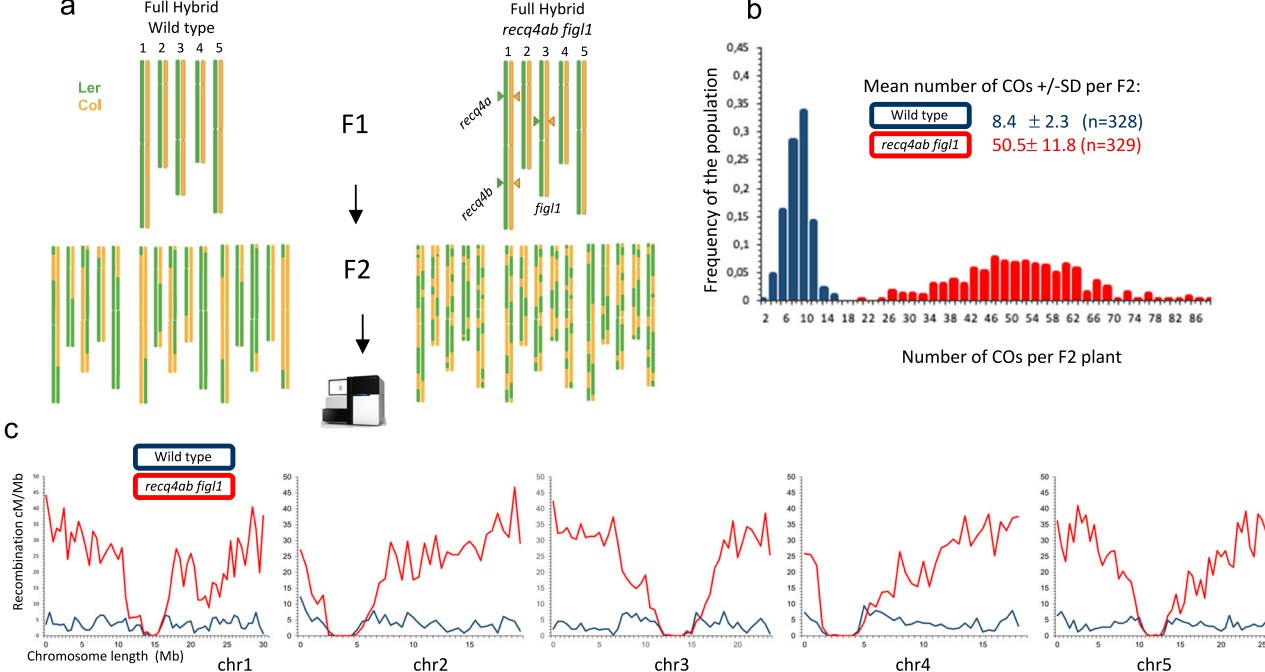

**Fig. 1 | Crossover frequencies are boosted in hybrid *recq4ab figl1*. a** Schematic representation of the full hybrid populations used in this study. **b** Distribution of crossover number per F2 plant. Wild type and *recq4ab figl1* are shown in dark blue and red, respectively. The mean crossover per F2 +/− standard deviation and the number of F2 plants analyzed (n) are indicated. **c** Distribution of crossovers along all five chromosomes in the hybrid population (*recq4ab figl1* in red vs. wild-type hybrids in dark blue), using non-overlapping 500 kb windows. Source data are provided as Source Data files and Supplementary Data 2.

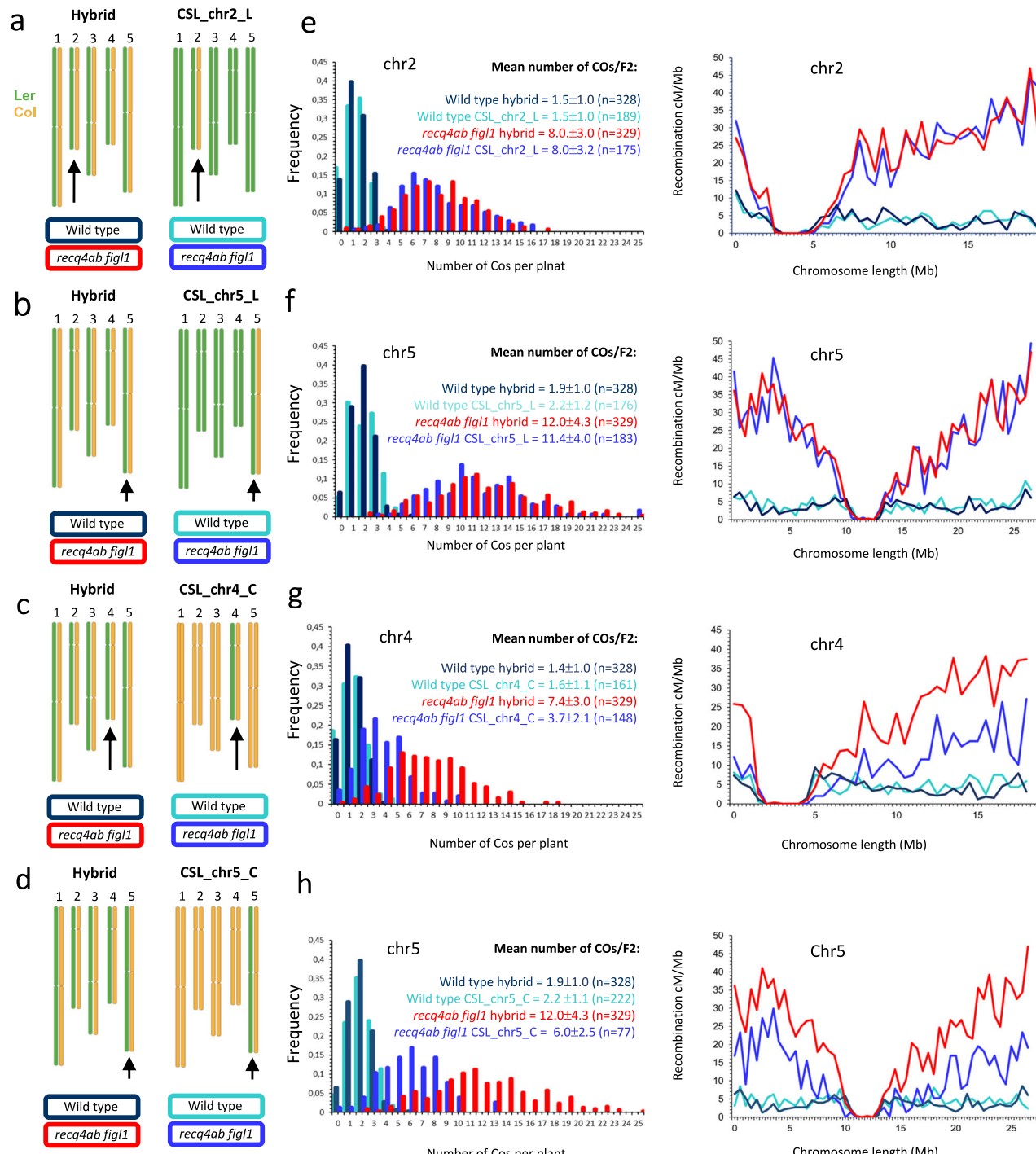

**Fig. 2 | *recq4ab figl1* boosts crossover frequencies in chromosome substitution lines. a–d** Schematic representation of the chromosome configuration of the populations analyzed in each plot. The analyzed chromosomes are marked by a black arrow. See also Figure S2. **e** Analysis of crossover frequency and distribution in CSL_chr2_L, in which chromosome 2 segregates and the other chromosomes are fixed Ler, compared to crossover frequency and distribution on chromosome 2 in full hybrid. **f** Analysis of crossover frequency and distribution in CSL_chr5_L, compared to crossover frequency and distribution on chromosome 5 in full hybrid. **g** Analysis of crossover frequency and distribution in CSL_chr4_C, in which chromosome 4 is segregating and the other chromosomes are fixed Col, compared to crossover frequency and distribution on chromosome 4 in full hybrid. **h** Analysis of crossover frequency and distribution in CSL_chr5_C, compared to crossover frequency and distribution on chromosome 5 in full hybrid. All analysis using non-overlapping 500 kb windows. Full hybrid data are from Fig. 1.

the rest of the genome is homozygous Col. In wild types, the crossover frequencies are similar in CSL_Chr4_C compared to full hybrids (Fig. 2g, h, 1.6 vs. 1.4; $p = 0.16$), and slightly increased in CSL_Chr5_C compared to full hybrids (2.2 vs. 1.9, $p = 0.002$). This suggests that a trans factor slightly affects the CO number on chromosome 5, with the Ler allele favoring COs in the wild type context. In contrast, chromosome 4 receives an average of 3.7

COs in *recq4ab figl1* CSL_Chr4_C, compared to 7.4 in *recq4ab figl1* full hybrid ($p < 10^{-6}$; Fig. 2g). Similarly, chromosome 5 receives on average 6 COs in *recq4ab figl1* CSL_Chr5_C, compared to 12 in *recq4ab figl1* full hybrid ($p < 10^{-6}$; Fig. 2h).

No aneuploids were detected in any of the wild-type F2 populations (total $n = 1183$). In *recq4ab figl1* F2 populations, no aneuploids were found

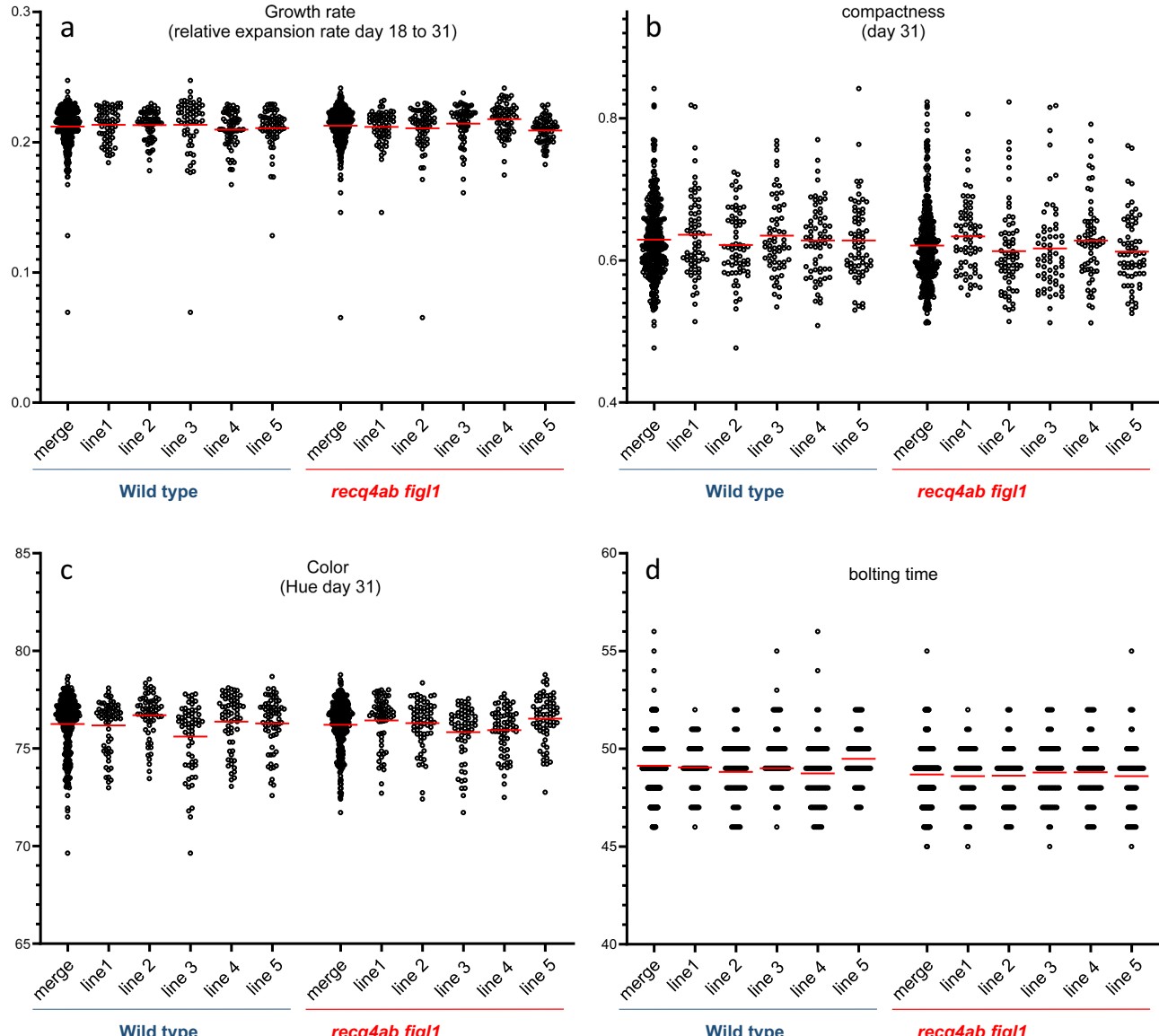

**Fig. 3 | Phenotypic distribution in wild-type *and recq4ab figl1* F2 populations.** For both wild-type and *recq4ab figl1*, the F2 progeny of five individual F1 plants (line) were scored for a range of phenotypes. The phenotypic value of each F2 population, and the merge, are shown. The red bars indicate the mean. **a** Relative expansion rate integrated between 18 and 31 Days After Sowing (DAS). **b** Compactness at 31 DAS. **c** Hue leaf color parameter at 31 DAS. **d** Bolting time.

for full hybrids (n = 329), CSL_chr2_L (n = 175) and CSL_chr5_L (n = 183). However one trisomic was found in the CSL_chr4_C population (n = 148) and one in the CSL_chr5_C (n = 77), both with an extra chromosome 2 (samples 3783_AM and 3786_U). This low frequency of aneuploidy observed in the r*ecq4ab figl1* progenies (2/912 = 0.2%), suggests that chromosome segregation at meiosis is not, or very marginally affected in this mutant.

### Hyper-recombination does not affect phenotypic variation

Taking advantage of the Phenoscope robots to carefully phenotype large cohorts of plants, we profiled the most interesting of these populations (full hybrid, CSL_chr2_L, and CSL_chr5_L) for diverse traits related to rosette growth and architecture, as well as phenology (time to bolting). Note that the scoring of time to bolting was done by hand and is, therefore, less resolutive -in time- and less accurate than the other image-based automated phenotypes. Figure 3a–d shows the distribution of relative growth rate, compactness, leaf color -Hue- and bolting time, comparing five repeats (F2 populations derived from five individual F1 plants) of wild-type and *recq4ab figl1* full hybrids. For each phenotype, the mean and variation of phenotypic

values are essentially indistinguishable between wild-type and *recq4ab figl1* F2 populations, showing that neither *recq4ab figl1* mutations nor elevated meiotic recombination affected the segregation of macroscopic traits.

### The resolution and power of QTL analysis are improved in hyper-recombinant segregating populations

Next we tested the effect of elevated COs on QTL mapping for the most heritable traits. Figure 4 presents the QTL maps for 3 traits (a. rosette compactness at day 31 after sowing, b. leaf color -hue- at day 31 after sowing, and c. time to bolting) as a comparison between wild-type and hyper-recombinant populations.

Overall, we detect very highly significant QTLs as well as mildly significant QTLs in all contexts. The analyzed traits appear to be essentially controlled by 2 QTLs with major and pleiotropic effects, one on chromosome 2 around position 60 and one on chromosome 5 around position 80. No clear effect of the recombination density on significance emerges, with typically comparable LOD Score values between populations for peaks at similar locations. In contrast, a striking effect of boosting recombination is observed on the resolution of the QTLs, resulting in much finer peaks with

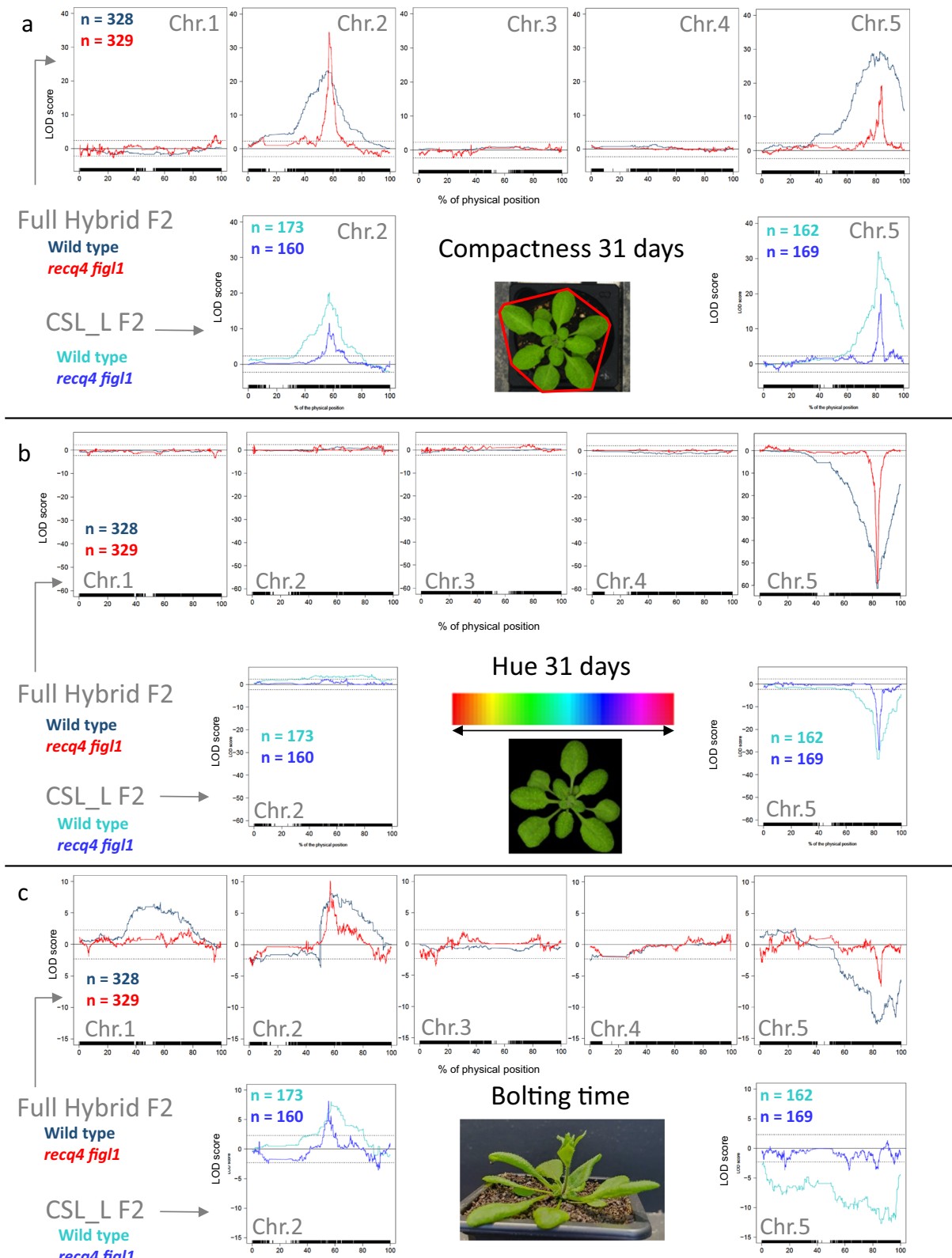

**Fig. 4 | QTL maps for three traits in six different populations.** Each graph presents LOD-score profiles computed by interval mapping for the indicated trait: (**a**) Compactness 31 days after sowing, (**b**) Hue 31 days after sowing and (**c**) bolting time). For each trait, the top row shows the QTL maps in full hybrids, comparing wild type with *figl1 recq4*; the bottom row shows the QTL maps in chromosome substitution lines (CSL_L), comparing wild type with *figl1 recq4*. "n" = number of plants analyzed in each population. The x-axes are in % of the physical position along the chromosome (Chr.). The y-axes represent the LOD-Score with additive allelic effect direction sign (+/−): a positive (negative) LOD Score indicates that the L*er* allele at this position increases (decreases) trait value with respect to the Col allele. Conservative LOD-score thresholds of +/−2.3LOD are represented as dashed lines.

significance dropping much more rapidly on each side of the peak. For instance, the most significant QTL (on chromosome 5 for Hue, full hybrid population; Fig. 4) reaches similar maximal significance in both recombination contexts but the extent of -for instance- a 3-LOD drop interval from the LOD maximum spans an interval 3 times bigger in a wild-type recombination context than in the hyper-recombinant population. Similarly, the peak beyond the significance threshold covers more than half of the chromosome in a classical recombinant population, while it is significant over ~15% of the chromosome in a hyper-recombinant background. A colocalizing QTL is mapped in CSL_chr5_L, essentially showing the same pattern, with a much finer peak in the hyper-recombinant lines. The significance for the likely same QTL is lower in the CSL_chr5_L compared to full hybrid, but it should be noted that the number of F2 plants analyzed is larger in the full hybrid sets compared to the CSL sets. Thus, in both full hybrid and CSLs, enhanced recombination resulted in a much finer definition of all the observed large-effect QTLs.

Another general trend is that the refined position of the LOD Score peaks in the hyper-recombinant context allows distinguishing smaller-effect QTLs in the vicinity of large-effect QTLs. For instance, at the bottom of chromosome 5, just south of a major peak, we detect another secondary QTL in the same allelic direction for Compactness and Hue traits (visible in the CSL_chr5_L population), but this one is masked by the shoulders of the main peak in the wild-type background. Similarly, on chromosome 2, a secondary QTL is detectable at position 40 for Compactness in the full hybrid population only in the hyper-recombinant background. For bolting time, the main peak on chromosome 2 seems to hide a QTL around position 90 in the opposite direction (typically a difficult situation to decompose), which is significant only in the hyper-recombinant full hybrid and CSL sets. An extreme and complex case arises when studying bolting time in the CSL_chr5_L set: chromosome 5 appears entirely above the significance threshold in the wild-type background, and this seems to be due to 3 independent QTLs in the same direction according to the hyper-recombinant analysis. The position of at least one of these QTLs is independently confirmed in the full hybrid population.

There are also some examples of significant peaks only detected in one of the backgrounds with no immediate explanation (e.g. the isolated peak at the bottom of chromosome 1 for compactness and hue is only significant in the hyper-recombinant population; the very wide peak on chromosome 1 for bolting time conversely only appears in the wild-type population, while the hyper-recombinant population does not clearly map anything, although there are multiple suggestive QTLs).

### QTL fine mapping resolution is enhanced in hyper recombinant lines vs. wild type and fine-mapping gives access to gene-scale resolution

As visible in Fig. 4, the confidence intervals of the QTL are dramatically reduced in the hyper-recombinant background. Because we encountered major-effect QTLs whose phenotypic effect could be discerned in a large proportion of the individual plants, we were able to proceed to fine-scale mapping of two of the QTLs aiming to identify the causal polymorphisms by zooming in and studying single recombination events together with the phenotype of the individual plants (Figs. 5 and 6).

Overlapping major QTL peaks are detected on chromosome 2 for compactness and flowering time. We used compactness for the fine mapping as the phenotypic scoring is more robust in our settings. Combining the wild-type full hybrid and CSL_Chr2_L populations, we could narrow the region of the QTL to an interval of 190 kb (chr2:11045-11235 kb) with one recombinant plant defining each of the left and right borders (Fig. 5). This region contained a few hundred protein-coding genes. When doing the same with recq4ab figl1 populations, and because of the much higher number of COs, we could reduce the interval to 16 kb (Chr2:11195-11212). The defined interval contains only four protein-coding genes (AT2G26300, AT2G26310, AT2G26320, and AT2G26330) (Fig. 5). Remarkably, one of them stands out as the possible causal gene, the receptor-like kinase *ERECTA* (*ER*) gene (AT2G26330), which is well known for being mutated

in L*er* and consequently segregating in our populations. The *ER* gene is involved in diverse processes regarding somatic development and immune response, but mainly, and as it concerns in this study, giving the specific leaf and rosette shape of the Landsberg *erecta* accession[32–34]. It is very likely that the *erecta* mutation (Chr2:11.209.133) is the causal variant under the chromosome 2 QTLs for compactness and flowering time.

Another major QTL was detected on chromosome 5, for compactness, Hue, and flowering time. We used compactness and Hue to fine-map the QTL, as they were both robust and coherent. Using the wild-type populations, the mapping defined an interval of 360 kb (Fig. 6), which contains hundreds of genes as possible candidates. In contrast, using the hyper recombinant *figl1 recq4* populations, we could narrow the interval down to 22 kb (Chr5:22608 to 22631). The defined interval contains five protein-coding genes (AT5G55860, AT5G55870, AT5G55880, AT5G55890, AT5G55893) and two transposons (AT5G55875 and ATG555896) (Fig. 6). One very tempting candidate is AT5G55860/*TREPH1*, whose mutation leads to modification of flowering time and rosette radius in response to mechanical stimulus[35]. This *TREPH1* gene contains two non-synonymous polymorphisms affecting the protein sequence in L*er* with respect to Col (Tair10 5:22611342 A > G, Thr376>A; 5:22611934 C > T, Ala>Val) that may affect the protein function.

## Discussion

Here we aimed to test the effect of enhanced recombination on QTL analysis. For this, we chose the *figl1 recq4ab* mutant, as it had the highest frequency of meiotic crossovers reported in plants[23]. While our previous measurement was performed with a set of 96 genetic markers, we used here an approach based on whole genome sequencing of progenies to analyze recombination in the same *recq4ab figl1* Col/Ler hybrid. We observed an average of 50.5 crossovers per *recq4ab figl1* F2 plant, to be compared with 8.4 in wild type. However, while the numbers are very similar in the wild type, the estimate in *recq4ab figl1* is lower than our previous one (3,037 cM, equivalent to 60 COs)[23]. We propose that the previous measurement was overestimated, as the consequence of the lower number of markers. Two phenomena may contribute to an overestimation when using a limited number of markers: First, genotyping errors become difficult to distinguish from real COs in a hyper-recombining context, where a double CO can be supported by a single marker. Second, some intervals reached large recombination frequencies leading to the necessity of using correction functions to estimate the genetic size (i.e. Haldane mapping function), taking into account missed multiple COs and potentially inflating the size of the map. The current approach is more reliable as it detects directly CO without the need for mathematical correction. However, we cannot exclude that a small number of very close double-COs can be missed, as we privileged a robust detection of COs (minimum double-CO distance = 150 kb, see methods). In any case, the new analysis confirms an exceptionally high frequency of crossovers in *recq4 figl1* compared to the wild type, opening the opportunity to test the effect of enhanced recombination on deciphering quantitative trait's genetics.

We also aimed to combine enhanced recombination with substitution lines, in which a single chromosome is segregating while the rest of the genome is fixed. In two populations, where either chromosome 2 or 5 segregates while the rest of the genome is fixed Ler, the number and distribution of COs are similar to the corresponding chromosome in the full hybrid context. Thus, CO frequency can be enhanced in substitution lines, exactly like in full hybrids. However, a different observation was made in two other lines, in which either chromosome 4 or 5 segregate, while the rest of the genome is homozygous Col: Wild type CO levels are similar (slightly higher) in these chromosomes compared to the same chromosome in full hybrids, but the CO levels in *recq4ab figl1* are reduced by half compared to the full hybrid. This still corresponds to a ~ 3-fold CO increase provoked by *recq4ab figl1* in these substitution lines, but this is less than the ~6-fold increase in the full hybrid or the increase in the two substitution lines in the Ler background. The weaker effect of *recq4ab figl1* on chromosomes 4 and 5 in the CSL_C context compared to the full hybrid suggests that a genetic factor acts

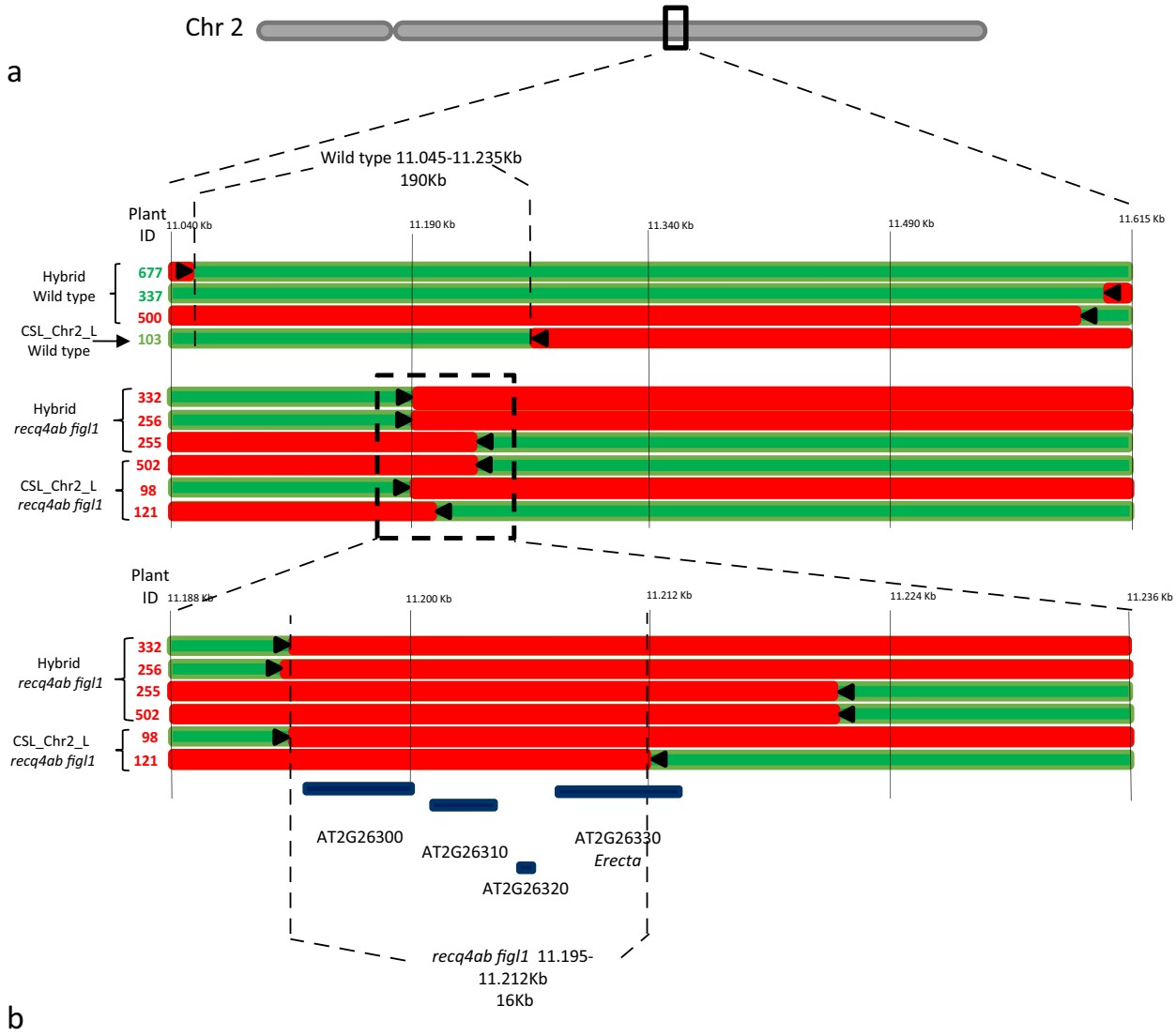

**Fig. 5 | Fine mapping of the major QTL detected on chromosome 2. a** Each line represents the genotype of a plant with a recombination point (crossover) in the region of interest (green = Ler, red = heterozygous Col/ler). The plant ID is shown on the left, with a color indicating its phenotypic class (in green the plants show a Landsberg-like phenotype and in red the plants show a Col/heterozygous-like phenotype - the Col allele is dominant). The direction in which the causal polymorphism is predicted to be is depicted with a black arrow. The most informative plants are shown for wild-type and *recq4 figl1* populations. The interval defined by the closest recombination points are indicated for wild-type and mutant populations. A zoom is made on the closest recombinants. **b** The positions of the crossovers are shown in the table, start and end defining the borders of the interval in which the crossover is detected.

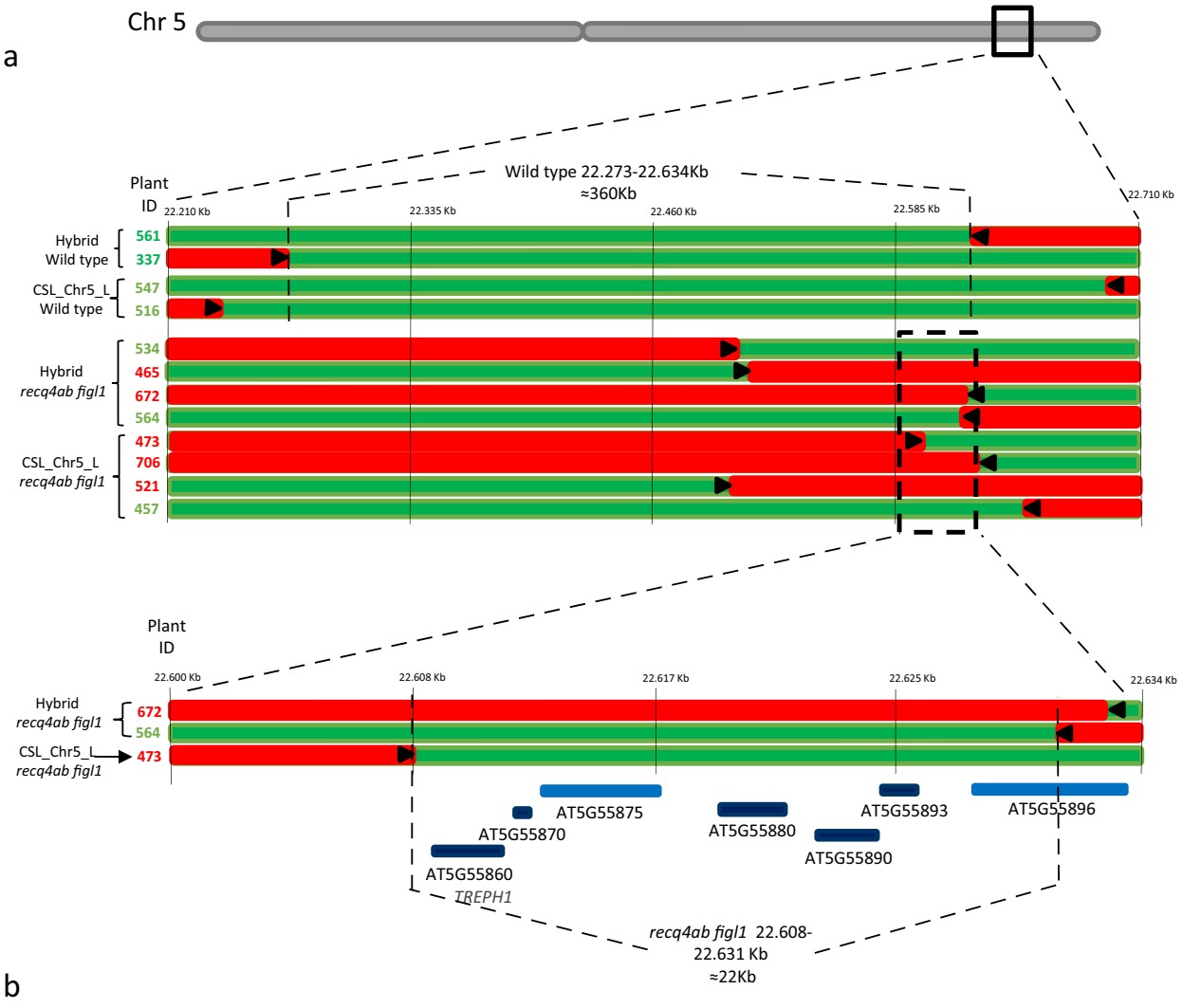

| plant ID | Population | CO start (Kb) | CO end (Kb) | Phenotype Compactness |
|---|---|---|---|---|
| 465 | Hybrid recq4ab figl1 | 22.514 | 22.514 | Col/Ler |
| 534 | Hybrid recq4ab figl1 | 22.502 | 22.503 | Landsberg |
| 672 | Hybrid recq4ab figl1 | 22.631 | 22.632 | Col/Ler |
| 564 | Hybrid recq4ab figl1 | 22.620 | 22.631 | Landsberg |
| 706 | CSL_Chr5_L recq4ab figl1 | 22.639 | 22.640 | Col/Ler |
| 473 | CSL_Chr5_L recq4ab figl1 | 22.608 | 22.609 | Landsberg |
| 521 | CSL_Chr5_L recq4ab figl1 | 22.500 | 22.502 | Col/Ler |
| 457 | CSL_Chr5_L recq4ab figl1 | 22.653 | 22.654 | Landsberg |
| 337 | Hybrid wild type | 22.273 | 22.280 | Landsberg |
| 561 | Hybrid wild type | 22.632 | 22.634 | Landsberg |
| 547 | CSL_Chr5_L wild type | 22.681 | 22.692 | Landsberg |
| 516 | CSL_Chr5_L wild type | 22.226 | 22.229 | Landsberg |

**Fig. 6 | Fine mapping of the major QTL detected on chromosome 5. a** Each line represents the genotype of a plant with a recombination point (crossover) in the region of interest (green = Ler, red = heterozygous Col/ler). The plant ID is shown on the left, with a color indicating its phenotypic class (in green the plants show a Landsberg-like phenotype, and in red the plants show a Col/heterozygous-like phenotype - the Col allele is dominant). The direction of where the causal

polymorphism is predicted to be is depicted with a black arrow. The most informative plants are shown for wild-type and *recq4 figl1* populations. A zoom is made on the closest recombinants. The interval defined by the closest recombination points are indicated for wild-type and mutant populations. **b** The positions of the crossovers are shown in the table, start and end defining the borders of the interval in which the crossover is detected.

in trans to modulate the effect of *recq4ab figl1* on recombination on the hybrid chromosome. The simplest model would involve a single factor situated on chromosome 1, 2, or 3, that would be polymorphic between Col and Ler, with the Ler allele favoring CO in a dominant manner. However, other scenarios are possible, and further experiments are needed to understand the basis of this observation.

Despite the large increase in recombination, *recq4ab figl1* hybrid mutants are fertile[23], and we detected a very low frequency of aneuploid in their progeny (0/329 for the full hybrid, and 2/583 for the CSL populations), suggesting that the high crossover rate does not impair chromosome segregation. In contrast, even a relatively mild defect in ensuring at least one crossover per chromosome at meiosis results in the production of aneuploids in the next generation[22,29,30]. Further, the F2 populations did not show any phenotypic defects compared to the wild type, with similar mean and variation in diverse phenotypes related to growth, shape, color, or flowering time (Fig. 3), further supporting the absence of genomic instability in *recq4 figl1*. It is perhaps not surprising to observe no specific consequences of enhanced recombination on phenotypic diversity of the F2 populations, as simulations predict an increased response to selection only after several generations of breeding[36,37]. Thus, we used segregating populations in which CO frequency is enhanced ~6-fold to test the effect on QTL mapping. The consequence of such an increase in recombination density is multiple, but it is the refined position of the LOD Score peaks which is the most spectacular: due to the increased recombination density with respect to the physical scale, the likelihood of a QTL drops extremely rapidly while we move away from the peak, as a consequence of the numerous lines harboring a recombination breakpoint in the neighboring intervals. The confidence interval of QTLs of diverse significance is therefore dramatically reduced in the hyper-recombinant background compared to the wild-type background, although the most likely localization of the major LOD peaks is usually not much changed. This is well expected and has been theorized both for QTL mapping resolution[38] and breeding efficiency[39–41]: here we demonstrate it in real segregating populations.

In the case of very-large effect loci -where the phenotype of an individual line can often be used to deduce its allelic state- this allowed us to perform a fine-mapping approach directly at the same step in the hyper-recombinant segregating population (without the need to develop an extra fine-mapping population), despite its reasonable size. We are showing that in such cases the hyper-recombinant context allows to propose candidate genes with a resolution close to that reached with association genetics in Arabidopsis.

In addition, the better resolution of major loci allows the detection of additional QTLs controlling non-negligible shares of the phenotypic variation, especially in cases when a large-effect QTL could be masking a smaller-effect QTL in its vicinity. Here, we have detected new loci next to major loci specifically in the hyper-recombinant population (this is expected due to the ability to observe more recombined allelic combinations between the two linked loci, which increases the ability to distinguish their respective contribution). In the most complex situations, a limiting resolution could ultimately result in false-negative QTL results (when two linked QTLs with opposite allelic effects cancel each other) or 'ghost' QTL (when two linked QTLs in the same allelic direction add up and generate a false peak in between).

We have demonstrated that QTL experiments are empowered by enhanced recombination. The increasing capability to manipulate recombination levels in various crop species[42–44] suggests that it will be possible to integrate this approach into plant breeding programs in the near future.

## Methods
### Genetic material
The following *Arabidopsis thaliana* mutants were used in this work (Figure S1): In Columbia-0: *recq4a-4* (GABI_203C07, N419423), *recq4b-2* (SALK_011357, N511130)[10] and *figl1-1*[16]. In Landsberg: *recq4a-W387X*[8] and *figl1-12*[16] (Figure S1). Wild type Col-0 and Ler-1 are 186AV1B4 and 213AV1B1 from the Versailles Arabidopsis stock center (https://publiclines.

versailles.inrae.fr/). Plants were grown in greenhouses or growth chambers (16 h day/8 h night, 20 °C). The library of parental homozygote CSLs in Columbia and in Landsberg background was described in[27].

The full hybrid populations were obtained by crossing Col *recq4a-4* +/− *recq4b-2* +/- *figl1-1* +/− as female with Ler *recq4a-W387X* +/- *figl1-12* +/− as male[23]. Among the obtained F1s, five plants with the wild-type genotype, and five plant carrying all the mutations were selected and selfed to contribute equally to the F2 populations, aiming to limit the bias of potential maternal phenotypic effects.

CSL_Chr4_C and CSL_Chr5_C populations were produced through successive crosses as illustrated in Figure S3 for CSL_Chr4_C: A first cross was done between *recq4ab figl1*+/− Columbia plants and parental homozygote CSLs with chromosome 4 and chromosomes 5 pure Landsberg, respectively (Figure S3A). The resulting plant was then backcrossed to the parental CSLs for chromosome 4 and chromosome 5 (Figure S3B). From the resultant BC1 plants, we selected the *recq4ab figl1* +/− and we identified the lines with pure Landsberg chromosome 4 (or chromosome 5, respectively), by using a set of 96 KASPAR markers as described in[23] (Figure S2C). The offspring from the selfing of the selected plant was crossed with Columbia *recq4ab figl1* +/− (Figure S2D). Among the resultant plants, we selected wild-type and *recq4ab figl1* individuals (S2E and F) as the parents of the populations used for recombination and QTL analyses.

For CSL_Chr2_L and CSL_Chr5_L, the same process was followed but using Landsberg *recq4ab figl1* +/− mutants and the CSL parental with Columbia chromosome 2 and chromosomes 5 in a Landsberg background. All mutations were genotyped twice for each generation. CSLs were genotyped in each generation with a few deletions/insertions markers polymorphic between Columbia and Landsberg (Supplementary Data 1)[45].

### Crossover analysis
The leaf samples of the selfing populations were used for DNA extraction and library preparation for Illumina sequencing[46] (~1 million reads, 2 x 150 bp) at the Max Planck Genome Center, Cologne, Germany (https://mpgc.mpipz.mpg.de/home/). Selected informative plants were sequenced deeper (an additional ~5 million reads) for the purpose of fine mapping. To create a list of high-quality SNP markers between Col and Ler, a strategy of combining the whole-genome alignment (syntenic SNPs identified by SyRI v1.2[47]) and short-read mapping (high-quality SNPs detected by inGAP-family[48]) was used[19,22,28]. The raw sequencing data was first checked by FastQC v0.11.9 (http://www.bioinformatics.babraham.ac.uk/projects/fastqc/), and then aligned to the TAIR10 Col reference genome[49,50] by BWA v0.7.17-r1188[51] with the default parameters. COs were detected using a sliding window-based method[19,22,28], with a window size of 50 kb and a sliding step of 25 kb. A double-CO must be supported by at least five windows, and a terminal CO by at least two windows. The identified COs were also manually and randomly checked by using inGAP-family. The list of CO positions can be found in Supplementary Data 2–4.

### Phenotyping
Several independent segregating F2 populations were phenotyped on the Phenoscope platform at INRAE-IJPB Versailles, France (https://phenoscope.versailles.inrae.fr/) using our standard protocol and control conditions (short days with 8 h photoperiod; days at 21 °C/65%RH; nights at 18 °C/65%RH; watering at 60% of the maximum soil water content) as in previous publications[52,53]. Traits related to rosette growth and architecture were acquired -as described previously- based on daily images during the vegetative stage. Here we focus on Compactness (the ratio between the projected rosette area and the area of the convex hull encompassing the rosette) and the Hue parameter of the average leaf color (expressed in the HSV scale), both acquired at 31 days after sowing (DAS); as well as the Relative Expansion Rate (RER18-31) integrated on the projected rosette area between 18 and 31 DAS. Bolting time was noted by hand every other day (on the same plants as above) as the time when the flower stem reaches 2 cm above the rosette. All phenotypic data are listed in Supplementary Data 5.

## QTL mapping

All QTL analyses were performed in R using the R/qtl package[54]. Classical interval mapping was performed for each trait in each population using the '*scanone*' function with default parameters to generate LOD Score statistics. The LOD Scores have been represented along each segregating chromosome using a scale expressed in % of the physical position of the marker along the chromosome. The LOD Score is marked with the sign of the allelic additive effect (additive effects are estimated as the phenotypic difference Ler - Col, using the '*effectscan*' function). All phenotypic and genotypic data are listed in Supplementary Data 5 (CC = homozygous Col, LL = homozygous Ler, CL = heterozygous Col/Ler).

## Statistics and reproducibility

Statistical analyses of QTLs were performed in R using the R/qtl package[54]. Tests for differences in CO frequencies are two-tailed Mann-Whitney tests performed in Prism 10.2.0. The number of samples and exact p values are provided. The population sizes of the F2 populations range from 77 (recq4ab figl1 CSL_chr5_C) to 329 (recq4ab figl1 hybrid).

## Reporting summary

Further information on research design is available in the Nature Portfolio Reporting Summary linked to this article.

## Data availability

DNA illumina sequence data have been deposited and can be downloaded from ebi.ac.uk/biostudies/arrayexpress (E-MTAB-13653 for full hybrid population, E-MTAB-13656 for Col CSL populations, E-MTAB-13657, for Ler CSL populations). All studied data are included in the article and/or supporting information. The source data behind the graphs in the paper can be found in Supplementary Data 6.

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

## Acknowledgements
This work was supported by the Rijk Zwaan Zaadteelt en Zaadhandel B.V to L.C.P., V.S., and R.M. This work has benefited from the support of IJPB's Plant Observatory technological platforms. The IJPB benefits from the support of Saclay Plant Sciences-SPS (ANR-17-EUR-0007). We thank the Max Planck-Genome-center Cologne (MP-GC) for DNA extraction, library preparation, and sequencing. This work was supported by core funding from the Max Planck Society and an Alexander von Humboldt Fellowship to Q.L.

## Author contributions
O.L and R.M. designed the research. LCP, EG, QL, MG, OL, and RM analyzed the data. V.S. and J.K. generated plant materials. B.H. supervised the whole-genome sequencing work. LCP, QL, O.L., and R.M. wrote the article with input from the other authors.

## Funding

## Competing interests
The authors declare the following competing interests: A Patent was filled by INRAe on the use of RECQ4 to manipulate meiotic recombination with RM listed among the inventors.
