## [Peer Review File · Communications Biology]

Reviewers' comments:

Reviewer #1 (Remarks to the Author):

Positional cloning of genes often relies on its fine genetic mapping through the identification of recombinant individuals between the markers used and the gene of interest. Finding recombinants depends on the recombination rate in the region where the gene is lying. Recombination events (crossovers) are rare and not evenly distributed along the genome with hot and cold spots of recombination. In their study, the authors compared the efficiency of Quantitative Trait Locus (QTL) detection in *Arabidopsis* (Columbia x Landsberg full hybrids and chromosome substitution lines (CSL) for chromosomes 2 and 5 (background Landsberg) and 4 and 5 (background Columbia)) with or without mutation for two genes enhancing recombination, *RecQ4* and *Figl1*. Comparative analyses in the full hybrids (*recq4ab figl1* vs WT) were consistent with those already observed by the same authors both in terms of crossover-rate increase and distribution of crossovers. Similar results are observed in CSLs when the background comes from Landsberg (CSL_Ch2_L and CSL_Ch5_L). Interestingly, the increase is twice lower when the background is from Columbia (CSL_Ch4_C and CSL_Ch5_C) which suggests the presence of additional trans factors that reduce the effect of the *recq4ab figl1* mutations. Boosting recombination rate allowed a better resolution of the QTLs for rosette compactness, leaf colour and time to bolting without reducing LOD-score values and with less plants. This also affects the sensitivity of the detection and gives the ability to observe additional QTLs with lower effect in the vicinity of the strongest ones. Moreover, the resolution achieved allows the identification of the candidate gene (*ERECTA* and *TREPH1*) for the shape of the rosette.

The paper is well written and shows convincing results with regards to the effect of the mutation of *RECQ4* and *FIGL1* on improving recombination rate (which is not new since it has largely been published by the same authors; e.g. Mieulet et al. 2018, <https://doi.org/10.1038/s41477-018-0311-x>; Seguela-Arnaud et al. 2015, www.pnas.org/cgi/doi/10.1073/pnas.1423107112) but also on the effect on QTL detection which is original but not surprising (since you increase simultaneously the number of markers and the recombination rate, you improve consistently the resolution of your QTLs). My main concern is about the trans effect of some factors that modulate the recombination rate, which for me is the most amazing part of the paper that would deserve to be more deeply explored and discussed. P5-6: Concerning chromosome 5 CSLs, it seems that the values between CSL_Ch5_L and CSL_Ch5_C are identical (2.2 COs) and significantly different from the WT (1.9 COs). In the first case (CSL_Ch5_L), a trans effect is mentioned but not in CSL_Ch5_C. Is there a reason why? If there is a trans effect in this case, it should be on Chr5 (cis effect?) since the rest of the genome is homozygous either for Ler or Col? Can you comment on that? In both cases there is a slight increase of COs in CSLs. Are there some genes on this chromosome that interact with *RECQ4* and/or *FIGL1* since this factor seems to slightly increase the number of crossovers? Or do you have evidence that it is involved in a different pathway?

P6: ..."a factor acts in trans to modulate the effect of *recq4ab figl1* on recombination on the hybrid chromosomes." Is it a factor coming from Ler which increases the number of crossovers or a factor coming from Col which reduces the number of crossovers? In that case, which one is dominant? If the factor is on Chr2 as you state in the discussion (P12), why don't you see any effect in CSL_2_L? Is it because there is no dosage effect? Is it because the Ler allele is dominant?

Based on this, there are two different points to elucidate and that need to be better discussed: (1) since in the WT context, you slightly increase the number of crossovers in both CSL_Ch5_L and CSL_Ch5_C, there should be two factors that interact on this chromosome, one coming from Ler and the other from Col. Since you have F2 individuals recombining on this chromosome only, you can use these to derive other segregating populations to clarify this; (2) since, when the background genome is from Ler, number of crossovers is much higher in the *recq4ab figl1* mutant, this means that Ler possesses a dominant gene/allele that is active only when the three genes are mutated. As you said, this gene probably locates on chromosomes 1, 2 or 3. It would be worth to test other accessions to see if Ler is the only one with such an effect. Also, using your current segregating *recq4ab figl1* F2 population, you could also select some individuals to derive other segregating populations.

P8: Since the QTL for the traits locate on Chr5, what would be the effect of using CSL_5_C instead of

CSL_5_L?

P14: "The capability to manipulate recombination levels in various crop species opens the possibility of integrating this approach into plant breeding programs." This should be toned down. The authors demonstrated that the mutation of FIGL1 can result in sterility in some species (Mieulet et al. 2018, <https://doi.org/10.1038/s41477-018-0311-x>). This is also the case for other genes like FANCM where reduced fertility is observed (Desjardin et al. 2022 <https://doi.org/10.1038/s41467-022-31438-6>). Recombination is increased but only in regions where it already recombines. Nothing is sure regarding regions that are blocked.

Reviewer #2 (Remarks to the Author):

This manuscript is the continuation of the work published in 2018 by the same research group (Fernandes et al. PNAS 115:2431-6). In the present work, genotyping was expanded using GBS by increasing the number of loci evaluated. Different F2 mapping populations of Arabidopsis were obtained using chromosome substitution lines (CSL) and combining *recq4ab/figl1* mutations by crossing. These mutations affect the expression of the RECQ4 and FIGL1 genes, so the mutants show an increased frequency of crossovers (CO). Comparison of the resulting F2 genotypes and QTL mapping with their WT counterpart showed that the frequency of CO increased dramatically (~6-fold), confirming previous results. Furthermore, *recq4ab/figl1* mutations improved the resolution and power of QTL analysis and did not affect the mapped phenotypes.

This manuscript showed an elegant experimental design and the results were convincing about the mechanism for enhancing QTL mapping by increasing recombination, induced by *recq4ab/figl1* mutations. Therefore, its publication in Communications Biology is recommended.

Reviewer #3 (Remarks to the Author):

The article by Capilla-Pérez et al clearly emphasises the importance of enhanced recombination in increasing the mapping resolution to a few candidate genes. This approach benefits from the power of traditional QTL mapping and the fine mapping of the GWAS. I found it of very high quality and interesting to researchers in both quantitative breeding and meiosis fields. However, I have some suggestions for improving the manuscript.

1. Regarding the two substitution populations, I wonder if the same level of cross-over enhancement will be observed if the background of the substitution lines changed, Col for chromosomes 2 or 5 and Ler for chromosomes 4 or 5. I believe this will add more insights in explaining why the enhancement is high only in the Ler background.

2. Related to point 1, is the enhancement related to specific chromosomes in specific backgrounds or more related to where the major genes controlling the traits are located?

Minor comments:

1. Few sentences discussing the findings in the results. These sentences should be moved to the discussion part or rewritten if they should appear in the results

a. Previous work combining knockouts for different anti-CO pathways showed that the genotype that gives the strongest increase so far, is obtained by mutating FIGL1 (AT3G27120) together with the two RECQ4 paralogs RECQ4A (AT1G10930) and RECQ4B (At1G60930) 23"

b. The observed CO frequency is slightly lower than our previous estimate 23 but confirms *recq4 figl1*

as the mutant with the highest recombination levels described to date (see also discussion)."

c. "These observations are consistent and refine previous conclusions 23."

d. "The weaker effect of *recq4ab figl1* on chromosomes 4 and 5 in the CSL_C context compared to the full hybrid suggests that a factor acts in trans to modulate the effect of *recq4ab figl1* on recombination on the hybrid chromosome."

e. "This low frequency of aneuploidy observed in the *recq4ab figl1* progenies ($2/912=0.2\%$), suggests that a very large number of COs does not, or very marginally, affect chromosome segregation at meiosis."

f. "The significance for the likely same QTL is lower in the CSL_chr5_L compared to full hybrid, which can be explained by at least two factors: [i] the number of F2 plants analyzed is larger in the full hybrid sets compared to the CSL sets; [ii] the epistatic context is different with other QTLs segregating in full hybrid but fixed Ler in the CSL. Thus, in both full hybrid and CSLs, enhanced recombination resulted in a much finer definition of all the observed large-effect QTLs."

reviewers' comments:

Reviewer #1 (Remarks to the Author):

Positional cloning of genes often relies on its fine genetic mapping through the identification of recombinant individuals between the markers used and the gene of interest. Finding recombinants depends on the recombination rate in the region where the gene is lying. Recombination events (crossovers) are rare and not evenly distributed along the genome with hot and cold spots of recombination. In their study, the authors compared the efficiency of Quantitative Trait Locus (QTL) detection in Arabidopsis (Columbia x Landsberg full hybrids and chromosome substitution lines (CSL) for chromosomes 2 and 5 (background Landsberg) and 4 and 5 (background Columbia)) with or without mutation for two genes enhancing recombination, RecQ4 and Figl1. Comparative analyses in the full hybrids (recq4ab figl1 vs WT) were consistent with those already observed by the same authors both in terms of crossover-rate increase and distribution of crossovers. Similar results are observed in CSLs when the background comes from Landsberg (CSL_Ch2_L and CSL_Ch5_L). Interestingly, the increase is twice lower when the background is from Columbia (CSL_Ch4_C and CSL_Ch5_C) which suggests the presence of additional trans factors that reduce the effect of the recq4ab figl1 mutations. Boosting recombination rate allowed a better resolution of the QTLs for rosette compactness, leaf colour and time to bolting without reducing LOD-score values and with less plants. This also affects the sensitivity of the detection and gives the ability to observe additional QTLs with lower effect in the vicinity of the strongest ones. Moreover, the resolution achieved allows the identification of the candidate gene (ERECTA and TREP1) for the shape of the rosette. The paper is well written and shows convincing results with regards to the effect of the mutation of RECQ4 and FIGL1 on improving recombination rate (which is not new since it has largely been published by the same authors; e.g. Mieulet et al. 2018, <https://doi.org/10.1038/s41477-018-0311-x>; Seguela-Arnaud et al. 2015, www.pnas.org/cgi/doi/10.1073/pnas.1423107112) but also on the effect on QTL detection which is original but not surprising (since you increase simultaneously the number of markers and the recombination rate, you improve consistently the resolution of your QTLs). My main concern is about the trans effect of some factors that modulate the recombination rate, which for me is the most amazing part of the paper that would deserve to be more deeply explored and discussed.

> Thank you for your positive, thorough, and constructive review. We agree that the effect of increased recombination on QTL mapping was rather expected, but this work represents, to our knowledge, the first experimental demonstration of this theoretical prediction. This is the main focus of the manuscript. We also agree that the observation of the lower recombination in certain CSL lines is very intriguing, and we describe it in detail. Understanding the genetic basis of this effect requires an amount of work which is well beyond the scope of this manuscript. However, as suggested below, we edited the text to discuss more the observations.

P5-6: Concerning chromosome 5 CSLs, it seems that the values between CSL_Ch5_L and CSL_Ch5_C are identical (2.2 COs) and significantly different from the WT (1.9 COs). In the first case (CSL_Ch5_L), a trans effect is mentioned but not in CSL_Ch5_C. Is there a reason why? If there is a trans effect in this case, it should be on Chr5 (cis effect?) since the rest of the genome is homozygous either for Ler or Col? Can you comment on that? In both cases there is a slight increase of COs in CSLs. Are there some genes on this chromosome that interact with RECQ4 and/or FIGL1 since this factor seems to slightly increase the number of crossovers? Or do you have evidence that it is involved in a different pathway?

> Indeed, both values are different from the wild type, as indicated in the manuscript ($p=0.002$ and 0.017 , respectively). We do not think that it could be a cis effect (a factor on chromosome 5) as chromosome 5 is fully heterozygous in the three genotypes (full hybrid, CSL_chr5_C, and CSL_chr5L). We agree that a trans effect involving a single gene is unlikely, as it is difficult to imagine a situation in which the highest value is observed in the col/col and the Ler/ler, with a lower value in the Col/Ler genotypes. One possibility is a trans effect involving two genes (not on chromosome 5), both with the positive allele being dominant. In that scenario, the full hybrid (+/-, +/-) would have more CO than Col (+/+, -/-) and Ler (-/-, +/+). This is speculative, and it is difficult at this stage to suggest what could be these factors. Identifying these factors might be also tedious as they act in the +/- 10% range. We have edited the manuscript to also mention the possibility of a trans effect in CSL-Chr5_C.

P6: ...”a factor acts in trans to modulate the effect of *recq4ab figl1* on recombination on the hybrid chromosomes.” Is it a factor coming from Ler which increases the number of crossovers or a factor coming from Col which reduces the number of crossovers? In that case, which one is dominant? If the factor is on Chr2 as you state in the discussion (P12), why don't you see any effect in CSL_2_L? Is it because there is no dosage effect? Is it because the Ler allele is dominant?

Based on this, there are two different points to elucidate and that need to be better discussed: (1) since in the WT context, you slightly increase the number of crossovers in both CSL_Chr5_L and CSL_Chr5_C, there should be two factors that interact on this chromosome, one coming from Ler and the other from Col. Since you have F2 individuals recombining on this chromosome only, you can use these to derive other segregating populations to clarify this; (2) since, when the background genome is from Ler, number of crossovers is much higher in the *recq4ab figl1* mutant, this means that Ler possesses a dominant gene/allele that is active only when the three genes are mutated. As you said, this gene probably locates on chromosomes 1, 2 or 3. It would be worth to test other accessions to see if Ler is the only one with such an effect. Also, using your current segregating *recq4ab figl1* F2 population, you could also select some individuals to derive other segregating populations.

P8: Since the QTL for the traits locate on Chr5, what would be the effect of using CSL_5_C instead of CSL_5_L?

As discussed above, we agree that there should be (at least) two factors, one from Col and one from Ler affecting recombination in the wild-type background, but we suggest that they are not on chromosome 5. The suggested experiments are of great interest but are beyond the scope of the present study. Concerning the potential trans factor(s) modifying CO in *recq4 figl1*, they are not necessarily the same as the factors modifying CO in wild-type, notably because the effects are in opposite directions (more CO in the CSL for wild-type, more CO in the hybrid for *recq4 figl1*). In the case of *recq4 figl1*, the simplest model is a single factor with a dominant allele favoring CO in Ler. We have edited the corresponding section of the discussion.

P14: “The capability to manipulate recombination levels in various crop species opens the possibility of integrating this approach into plant breeding programs.” This should be toned down. The authors demonstrated that the mutation of FIGL1 can result in sterility in some species (Mieulet et al. 2018, <https://doi.org/10.1038/s41477-018-0311-x>). This is also the case for other genes like FANCM where reduced fertility is observed (Desjardin et al. 2022 <https://doi.org/10.1038/s41467-022-31438-6>). Recombination is increased but only in regions where it already recombines. Nothing is sure regarding regions that are blocked.

> We agree that FIGL1 and some other genes that increase recombination in arabidopsis are not suitable for increasing recombination in crops. However, other genes are suitable. Indeed, nothing is sure about regions that are blocked, but in this manuscript, we showed that increasing

recombination is useful, even without unlocking regions. An increase in recombination was obtained in crop species, but not (yet) to the levels observed in the *Arabidopsis figl recq4*. We have thus toned down this statement in the revised manuscript.

Reviewer #2 (Remarks to the Author):

This manuscript is the continuation of the work published in 2018 by the same research group (Fernandes et al. PNAS 115:2431-6). In the present work, genotyping was expanded using GBS by increasing the number of loci evaluated. Different F2 mapping populations of *Arabidopsis* were obtained using chromosome substitution lines (CSL) and combining *recq4ab/figl1* mutations by crossing. These mutations affect the expression of the *RECQ4* and *FIGL1* genes, so the mutants show an increased frequency of crossovers (CO). Comparison of the resulting F2 genotypes and QTL mapping with their WT counterpart showed that the frequency of CO increased dramatically (~6-fold), confirming previous results. Furthermore, *recq4ab/figl1* mutations improved the resolution and power of QTL analysis and did not affect the mapped phenotypes.

This manuscript showed an elegant experimental design and the results were convincing about the mechanism for enhancing QTL mapping by increasing recombination, induced by *recq4ab/figl1* mutations. Therefore, its publication in *Communications Biology* is recommended.

> Thank you for your positive review. We are pleased that you appreciated the work.

Reviewer #3 (Remarks to the Author):

The article by Capilla-Pérez et al clearly emphasises the importance of enhanced recombination in increasing the mapping resolution to a few candidate genes. This approach benefits from the power of traditional QTL mapping and the fine mapping of the GWAS. I found it of very high quality and interesting to researchers in both quantitative breeding and meiosis fields. However, I have some suggestions for improving the manuscript.

> Thank you for your positive review. We are pleased that you appreciated the work.

1. Regarding the two substitution populations, I wonder if the same level of cross-over enhancement will be observed if the background of the substitution lines changed, Col for chromosomes 2 or 5 and Ler for chromosomes 4 or 5. I believe this will add more insights in explaining why the enhancement is high only in the Ler background.

2. Related to point 1, is the enhancement related to specific chromosomes in specific backgrounds or more related to where the major genes controlling the traits are located?

> The suggested experiments are of great interest but represent a substantial amount of work. The main aim of this study was to test the effect of enhanced recombination on QTL mapping. Including more SLs would contribute little to that question. Including more populations, like the experiments suggested by reviewer #1 would help to understand the unexpected behavior of some SLs, but this is beyond the scope of the present study.

Minor comments:

1. Few sentences discussing the findings in the results. These sentences should be moved to the discussion part or rewritten if they should appear in the results

a. Previous work combining knockouts for different anti-CO pathways showed that the genotype that gives the strongest increase so far, is obtained by mutating FIGL1 (AT3G27120) together with the two RECQ4 paralogs RECQ4A (AT1G10930) and RECQ4B (At1G60930) 23"

b. The observed CO frequency is slightly lower than our previous estimate 23 but confirms recq4 figl1 as the mutant with the highest recombination levels described to date (see also discussion)."

c. "These observations are consistent and refine previous conclusions 23."

d. "The weaker effect of recq4ab figl1 on chromosomes 4 and 5 in the CSL_C context compared to the full hybrid suggests that a factor acts in trans to modulate the effect of recq4ab figl1 on recombination on the hybrid chromosome."

e. "This low frequency of aneuploidy observed in the recq4ab figl1 progenies ($2/912=0.2\%$), suggests that a very large number of COs does not, or very marginally, affect chromosome segregation at meiosis."

f. "The significance for the likely same QTL is lower in the CSL_chr5_L compared to full hybrid, which can be explained by at least two factors: [i] the number of F2 plants analyzed is larger in the full hybrid sets compared to the CSL sets; [ii] the epistatic context is different with other QTLs segregating in full hybrid but fixed Ler in the CSL. Thus, in both full hybrid and CSLs, enhanced recombination resulted in a much finer definition of all the observed large-effect QTLs."

> We have edited the text according to your suggestions. However, in the specific case a. we prefer to keep the information in the result section as we believe it helps the reader to understand the experiments.

REVIEWERS' COMMENTS:

Reviewer #1 (Remarks to the Author):

I found that the revised version of the paper brings some clarifications regarding questioning that the first version raised. I understand that additional experiments will delay the publication too much, even if they would contribute to better clarify some interesting points.

Reviewer #3 (Remarks to the Author):

I fully understand that adding additional experiments is time-consuming, and I agree with the authors that the work, in its present form, is informative enough to be published.